

# Early haemodynamic predictors of poor functional outcomes in patients with acute ischaemic stroke receiving endovascular therapy: a single-centre retrospective study in China

Yanyan Hu[1], Shizhong Zhang[2], Jiajun Zhang[3], Xin Wang[4], Feng Zhang[4], Hong Cui[4], Hui Yuan[1] and Wei Zheng[5]

[1] Department of Neurology, The Second Affiliation Hospital of Shandong First Medical University, Taian, Shandong, China
[2] Department of Neurosurgery, The Affiliated Taian City Central Hospital of Qingdao University, Taian, Shandong, China
[3] Department of Ultrasound, The Second Affiliation Hospital of Shandong First Medical University, Taian, Shandong, China
[4] Department of Ultrasound, The Affiliated Taian City Central Hospital of Qingdao University, Taian, Shandong, China
[5] Department of Neurosurgery, The Second Affiliation Hospital of Shandong First Medical University, Taian, Shandong, China

Corresponding authors
Hui Yuan, yuanhui0314@163.com
Wei Zheng, zwzw1001@126.com

## ABSTRACT

**Background.** Changes in cerebral haemodynamics following endovascular therapy (EVT) for large-vessel occlusion stroke may affect the outcomes of patients with acute ischaemic stroke (AIS); however, evidence supporting this belief is limited. This study aims to identify the early haemodynamic predictors of poor outcomes in patients with AIS caused by anterior circulation large-artery occlusion after undergoing EVT and to evaluate the usefulness of these indicators in predicting functional outcomes at 90 days.
**Methods.** This retrospective study was conducted at a single academic hospital, using prospectively collected data. We enrolled adult patients with acute anterior circulation stroke who underwent EVT. Transcranial colour-coded sonography (TCCS) examinations of the recanalised and contralateral middle cerebral artery (MCA) were performed within 12 h after undergoing EVT. Haemodynamic indicators were analysed to determine their association with poor functional outcomes (modified Rankin Scale: 3–6) 90 days after stroke. Receiver operating characteristic (ROC) curves were used to evaluate the usefulness of haemodynamic indicators in predicting functional outcomes.
**Results.** In total, 108 patients (median age: 66 years; 69.4% males) were enrolled in this study. Complete recanalization was achieved in 93 patients (86.1%); however, 60 patients (55.6%) had a poor 90-day outcome. The peak systolic velocity (PSV) ratio, adjusted PSV ratio, mean flow velocity (MFV) ratio, and adjusted MFV ratio of the MCA were significantly higher in patients with poor prognosis than in patients with good prognosis ($p < 0.02$). A multivariate logistic regression analysis showed that higher PSV ratio, adjusted PSV ratio, MFV ratio, and adjusted MFV ratio were independently associated with a poor 90-day outcomes (adjusted odds ratio: 1.11–1.48 for every 0.1 increase; $p < 0.03$). Furthermore, adding the adjusted MFV ratio significantly improved

the prediction ability of the basic model for the 90-day poor functional outcome using the ROC analysis, the areas under ROC curves increased from 0.75 to 0.85 ($p = 0.013$).

**Conclusions.** Early TCCS examination may help in predicting poor functional outcomes at 90 days in patients with AIS who underwent EVT. Moreover, combining novel TCCS indicators (adjusted MFV ratio) with conventional parameters improved the prediction ability of the base model.

## INTRODUCTION

Endovascular therapy (EVT) has proven effectiveness for patients with acute ischaemic stroke (AIS) (*Goyal et al., 2016*). With the growing application of EVT, interest in understanding the cerebral haemodynamic changes that occur after performing EVT has risen.

In the acute phase following reperfusion, wide variations show in the cerebral haemodynamic patterns of patients with stroke (*Patel et al., 2013*; *Powers et al., 2018*). These variations may result in different outcomes after reperfusion therapy (*Shahripour et al., 2021*; *Yu et al., 2018*). Therefore, assessment after recanalization is important to early identify high-risk patients, to reduce the risk of complications, and to improve clinical outcomes in patients with haemodynamic disturbances through intervention.

To assess cerebral haemodynamics in patients with AIS, a safe and valid imaging modality is required to measure these indicators in real-time and track the relevant dynamic changes at bedside (*Shahripour et al., 2021*). Although imaging modalities such as magnetic resonance imaging (MRI), computed tomography (CT)/CT perfusion (CTP), and digital subtraction angiography play a role in the management of ischaemic stroke (*Zhang et al., 2019*), transcranial colour-coded sonography (TCCS), which is widely available and affordable in both in- and out-patient visits, provides unique real-time insights into haemodynamic findings without the use of ionising radiation or invasive procedures (*Wu et al., 2022*). Adequate characterisation and understanding of post-EVT cerebral haemodynamics measured using TCCS can be clinically useful. Follow-up examinations can help in identifying patients at potential risk, and identification of TCCS indicators for risk stratification may support decision-making in patient care and treatment.

In this retrospective study, we aim to investigate the changes in post-EVT cerebral haemodynamics in patients with AIS using TCCS, to clarify the association between haemodynamic indicators and functional outcomes 90 days after undergoing EVT.

## METHODS

### Ethical approval

This study was approved by the Ethics Committee of the Second Affiliated Hospital of Shandong First Medical University (approval number: 2021-098) and conducted in

accordance with the Declaration of Helsinki as revised in 2013. A written informed consent was obtained from conscious patients or the legal representatives of unconscious patients.

## Study population and therapeutic process

We identified all consecutive hospitalised patients who underwent EVT for acute large-artery occlusion (LAO) in the anterior circulation (intracranial internal carotid artery (ICA) or middle cerebral artery (MCA)) between October 2021 and September 2022 at our stroke centre (the Second Affiliated Hospital of Shandong First Medical University). All patients underwent a CT and/or CTP or an MRI examination before treatment to confirm the diagnosis.

Interventional neurologists performed mechanical thrombectomy using stent retrievers or clot aspiration systems. The treatment results were evaluated using the modified treatment in cerebral infarction (mTICI) score at the end of the EVT. mTICI grades 2b and 3 indicated complete recanalization, and mTICI grade 2 indicated partial recanalization (*Wang et al., 2015*). After undergoing EVT, all patients were treated in the neurointensive care unit. The target blood pressure level was <180/105 mmHg according to international guideline recommendations (*Jauch et al., 2013*).

We excluded patients with $\geq$ 70% stenosis of the extracranial ICA or with $\geq$ 50% stenosis of the contralateral MCA. Moreover, patients with minimal recanalization (mTICI: 1) or no recanalization (mTICI: 0) of the lesion-side MCA were excluded because cerebral blood flow indicators could not be obtained. Additionally, we excluded patients who did not complete the TCCS assessment within 12 h after undergoing EVT and those without available temporal windows or complete clinical data during the study period.

## Clinical data collection

Patient demographics and clinical variables were collected and analysed. The clinical variables included major vascular risk factors, stroke subtype, stroke severity, and treatment. Major vascular risk factors included hypertension, hyperlipidaemia, diabetes mellitus, coronary artery disease, atrial fibrillation, smoking, alcohol use, previous stroke, and medication use. Stroke subtypes were classified according to the Trial of Org 10172 in Acute Stroke Treatment criteria (*Adams Jr et al., 1993*). Stroke severity was assessed using National Institutes of Health Stroke Scale (NIHSS) scores at admission and Alberta Stroke Program Early CT (ASPECT) scores of the initial brain CT images. Regarding treatment, the use of intravenous recombinant tissue plasminogen activators and EVT parameters such as time from onset to treatment, time from onset to reperfusion, and type of EVT were recorded.

## TCCS

All patients were evaluated using TCCS (devices: Mindray M9, handheld transducers SP5-1S, 1–5 MHz; Mindray Biomedical Electronics Co., Ltd., Shenzhen, Guangdong, China) within 12 h of undergoing EVT by three professional ultrasound specialists blinded to the clinical data. To ensure an accurate comparison of the bilateral MCA blood flow velocities (BFVs), TCCS monitoring was conducted at a depth of 45–60 mm. Gray and colour gains were adjusted for each patient to obtain optimal image quality. Most patients

opted not to use ultrasound contrast agents due to concerns about cost and potential negative consequences. Therefore, we did not use ultrasound contrast agents during TCCS monitoring.

During examination, we recorded haemodynamic data, including peak systolic velocity (PSV), end-diastolic velocity (EDV), mean blood flow velocity (MFV), and pulsatility index (PI), from the MCA to assess the vessel status and cerebral haemodynamics. The highest BFV value was recorded after correcting for the insonation angle. The contralateral artery was examined along with the treated vessel. Moreover, we recorded the patients' blood pressure at the time of ultrasonography and the time interval between undergoing EVT and TCCS.

To account for inter-individual differences (*Kneihsl et al., 2022*), the MCA-MFV ratio (ipsilateral MFV (iMFV)/contralateral MFV (cMFV)) and PSV ratio (ipsilateral PSV (iPSV)/contralateral PSV (cPSV)) were calculated. To evaluate the impact of differences in BFV in the bilateral MCA on prognosis, we introduced the adjusted PSV ratio, which was calculated as follows: If iPSV ≥ cPSV, the adjusted PSV ratio was equal to the PSV ratio. In contrast, if iPSV <cPSV, then the adjusted PSV ratio was equal to the inverse of the PSV ratio. The adjusted MFV ratio was calculated similarly.

## Follow-up

Whenever possible, follow-up information was obtained at 90-day personal clinical visits. For the remaining patients, the clinical outcomes were assessed *via* telephone calls with the patient or the relatives. The modified Rankin Scale (mRS) score was used to estimate patient outcomes; a poor functional outcome was defined as an mRS score of 3–6 (*Kneihsl et al., 2018*).

## Statistical analysis

Paired continuous variables were analysed using the Wilcoxon matched-pairs signed-rank test, and unpaired groups were analysed using the Student's $t$-test or Mann–Whitney $U$ test. The categorical variables were analysed using the $\chi^2$ test or Fisher's exact test. Multivariate logistic regression analysis was used to evaluate the association with poor functional outcomes by calculating the corresponding odds ratios (ORs) and 95% confidence intervals (CIs), and only those factors that had statistical significance with univariate logistic regression analysis ($p < 0.05$) were employed. Statistical significance was set at a $p$-value of <0.05.

The optimum cut-off values of haemodynamic predictors, which were significantly associated with poor functional outcomes, were calculated using receiver operating characteristic (ROC) analysis. Furthermore, by comparing the ROC curves, we investigated whether adding haemodynamic predictors could improve the prediction ability of the regression model for poor functional outcomes at 90 days in AIS patients undergoing EVT. A two-tailed $p$-value of <0.05 was considered statistically significant. Statistical analyses were performed using the SPSS software (version 22.0; SPSS Inc., Chicago, IL, USA) and Graphpad Prism (version 8.3.0; Graphpad Software, San Diego, CA, USA).

## RESULTS

### Population characteristics

During the study period, 156 consecutive patients with AIS caused by LAO of the anterior circulation were hospitalised. However, 48 patients were excluded as they did not meet the inclusion criteria. A detailed flowchart outlining the inclusion and exclusion criteria is shown in Fig. 1. The analysis was conducted on 108 patients (median age: 66 years; 69.4% men). Of these patients, 63 (58.3%) had an MCA occlusion, 31 (28.7%) had an intracranial ICA occlusion, and 14 (13.0%) had occlusions in both the intracranial ICA and MCA. The median NIHSS score at admission was 18 (interquartile range (IQR): 14–22). Before undergoing EVT, 41 patients (38.0%) underwent intravenous thrombolysis. Complete recanalization (mTICI: grade 3) was achieved in 93 patients (86.1%), and partial recanalization (mTICI: grade 2) was achieved in 15 (13.9%). After undergoing EVT, 18 patients experienced intracranial haemorrhage, 12 showed an increase in the infarct size, and two experienced diffuse cerebral oedema. Of the total 108 patients, 60 (55.6%) had poor 90-day outcomes, and 90-day all-cause mortality was 13.0%. Deaths were attributed to tentorial herniation caused by cerebral haemorrhage or diffuse oedema (in nine patients) and pulmonary infection (in five patients). A complete summary of the demographic and clinical characteristics of the study population is presented in Table 1.

### TCCS indicators of the ipsilateral MCA (iMCA) and the contralateral MCA (cMCA)

TCCS was performed at a median time interval of 3.8 h after undergoing EVT. The median of iPSV, iEDV, iMFV, and iPI were 164.4 (IQR: 110.4–215.5) cm/s, 60.2 (IQR: 45.0–86.1) cm/s, 98.1 (IQR: 68.7–121.2) cm/s, and 1.01 (IQR 0.84–1.18), respectively; the median of cPSV, cEDV, cMFV, and cPI were 107.4 (IQR: 95.5–128.0) cm/s, 40.7 (IQR: 34.8–49.9) cm/s, 68.5 (IQR: 60.1–80.8) cm/s, and 0.92 (IQR: 0.80–1.06), respectively. Our results showed that PSV, EDV, MFV, and PI were significantly higher in the iMCA than in the cMCA, and the differences were statistically significant ($Z = -7.53$, $-7.20$, $-6.53$, and $-2.29$, respectively; $p < 0.001$, 0.001, 0.001, and $= 0.022$, respectively) (Fig. 2).

### Clinical and TCCS characteristics of patients with poor functional outcomes at 90 days

In the univariate logistic regression analysis, the group with poor functional outcomes was of older age (median [IQR]: 69 [58.8–73.8] years vs. 64 [58–69.5] years; $p = 0.035$), had higher admission NIHSS scores (median [IQR]: 20 [16–23] vs. 17 [9.3–19.8], $p < 0.001$), and lower ASPECT scores (median [IQR]: 8 [8–9] vs. 9 [8−9.8], $p < 0.001$). Multivariate logistic regression analysis showed that, after adjustment, the initial NIHSS scores (adjusted OR (aOR): 1.07; 95% CI [1.01–1.13]; $p = 0.023$) and ASPECT scores (aOR: 0.53; 95% CI [0.32–0.86]; $p = 0.010$) were still independently associated with functional outcomes (Table 2).

To investigate the effect of haemodynamics on patient outcomes after undergoing EVT, we analysed the correlation between TCCS variables and 90-day outcomes. Patients with poor outcomes had significantly higher PSV ratio (mean ± standard deviation (SD):

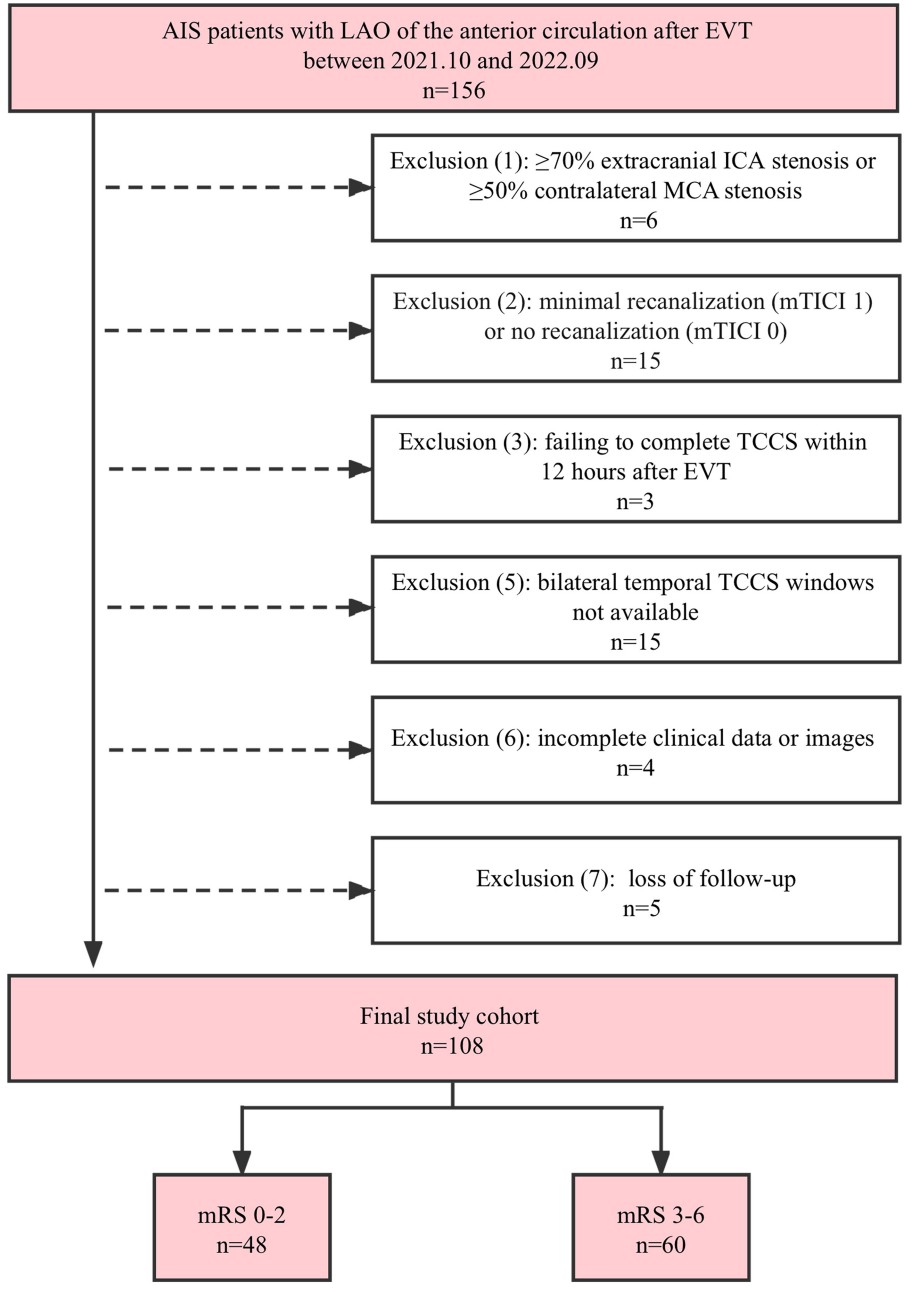

**Figure 1 Flowchart of the study population.** AIS, acute ischemic stroke; EVT, endovascular therapy; ICA, internal carotid artery; LAO, large artery occlusion; MCA, middle cerebral artery; mRS, modified Rankin Scale; mTICI, modified Thrombolysis in Cerebral Infarction; TCCS, transcranial color-coded sonography.

$1.60 \pm 0.64$ *vs.* $1.35 \pm 0.31$; $p = 0.016$), higher adjusted PSV ratio (median [IQR]: 1.75 [1.38–1.99] *vs.* 1.37 [1.09–1.61]; $p < 0.001$), higher MFV ratio (mean $\pm$ SD: $1.48 \pm 0.64$ *vs.* $1.22 \pm 0.25$; $p = 0.010$), and higher adjusted MFV ratio (median [IQR]: 1.60 [1.27–1.86] *vs.* 1.24 [1.11–1.41]; $p < 0.001$) than those of the good outcome group. Furthermore,
**Table 1  Baseline characteristics of 108 patients who underwent EVT for recanalization of anterior circulation LAO.**

| Variables | Total patients (*n* = 108) |
|---|---|
| **Demographics** | |
| Age, years, median (IQR) | 66 (58–72) |
| Male, n (%) | 75 (69.4) |
| **Medical history, n (%)** | |
| Hypertension | 72 (66.7) |
| Hyperlipidemia | 34 (31.5) |
| Diabetes mellitus | 23 (21.3) |
| Coronary artery disease | 35 (32.4) |
| Atrial fibrillation | 27 (25.0) |
| Smoking | 50 (46.3) |
| Alcohol | 49 (45.4) |
| Previous stroke | 20 (18.5) |
| Antiplatelet therapy/oral anticoagulation | 15 (13.9) |
| **Stroke mechanism, n (%)** | |
| LAA | 61 (56.5) |
| Cardioembolism | 30 (27.8) |
| Other etiologies | 17 (15.7) |
| **Admission NIHSS, median (IQR)** | 18 (14–22) |
| **ASPECTS, median (IQR)** | 9 (8–9) |
| **Procedure timings** | |
| Onset-to-puncture time, min, median (IQR) | 256 (185.5–310.5) |
| Onset-to-reperfusion time, min, median (IQR) | 350 (302–425) |
| Reperfusion-to-TCCS, min, median (IQR) | 229.5 (184.5–303.5) |
| **MBP during TCCS monitoring, mean ± SD, mmHg** | 99.2 ± 13.76 |
| **Occlusion artery, n (%)** | |
| ICA + MCA | 14 (13.0) |
| ICA | 31 (28.7) |
| MCA | 63 (58.3) |
| **Acute stroke therapy, n (%)** | |
| Bridging thrombolysis | 41 (38.0) |
| Intra-arterial thrombolysis | 19 (17.6) |
| Stent retriever | 87 (80.6) |
| Aspiration | 11 (10.2) |
| Angioplasty or stent | 35 (32.4) |
| **mTICI, n (%)** | |
| 2a | 8 (7.4) |
| 2b | 7 (6.5) |
| 3 | 93 (86.1) |

**Table 1** (*continued*)

| Variables | Total patients (*n* = 108) |
|---|---|
| **mRS score at 90-day, n (%)** | |
| 0–2 | 48 (44.4) |
| 3–5 | 46 (42.6) |
| 6 | 14 (13.0) |

**Notes.**

ASPECTS, Alberta Stroke Program Early Computed Tomography Score; EVT, endovascular therapy; ICA, internal carotid artery; IQR, interquartile range; LAA, large artery atherosclerosis; LAO, large artery occlusion; MBP, mean blood pressure; MCA, middle cerebral artery; mRS, modified Rankin Scale; mTICI, the modified treatment in cerebral infarction; NIHSS, National Institute of Health Stroke Scale; SD, standard deviation; TCCS, transcranial color-coded sonography.

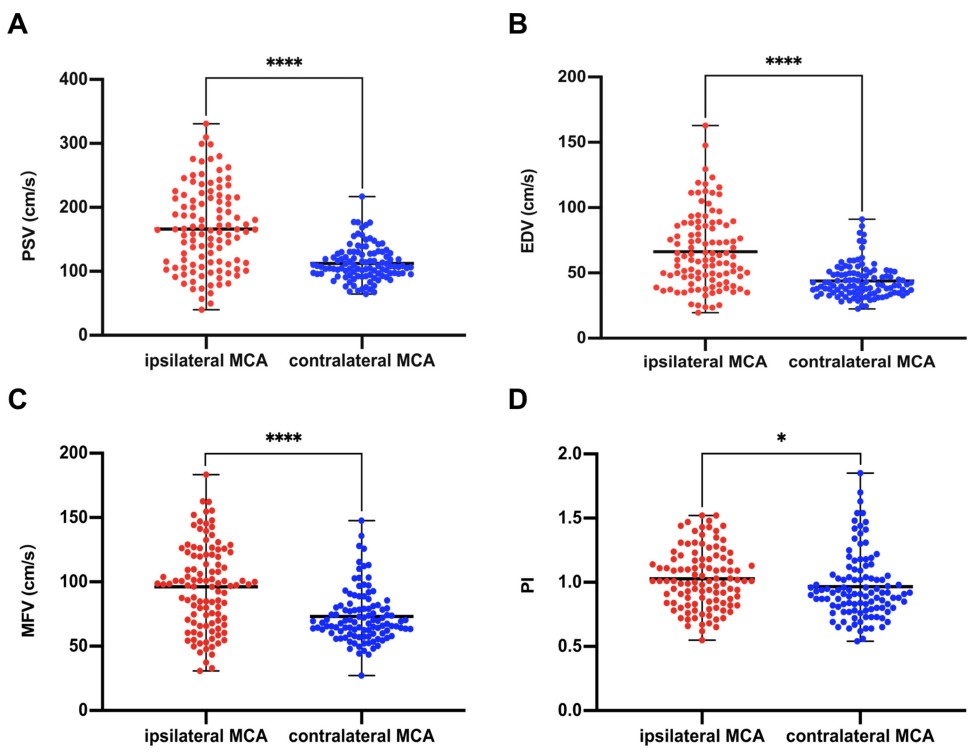

**Figure 2 Comparison of ultrasound parameters between ipsilateral (recanalized) and contralateral MCA.** PSV (A), EDV (B), MFV (C), PI (D) in the ipsilateral MCA were higher than those in the contralateral side (PSV, $p < 0.0001$; EDV, $p < 0.0001$; MFV, $p < 0.0001$; PI, $p = 0.02$). ****, $p < 0.0001$; *, $p < 0.05$. EDV, end diastolic velocity; MCA, middle cerebral artery; MFV, mean flow velocity; PI, pulsatility index; PSV, peak systolic velocity.

after adjusting for the initial NIHSS and ASPECT score, an increased PSV ratio (aOR: 1.11 for every 0.1 increase; 95% CI [1.02–1.22]; $p = 0.022$), an increased adjusted PSV ratio (aOR: 1.31 for every 0.1 increase; 95% CI [1.13–1.50]; $p < 0.001$), an increased MFV ratio (aOR: 1.15 for every 0.1 increase; 95% CI [1.04–1.28]; $p = 0.007$), and an increased adjusted MFV ratio (aOR: 1.48 for every 0.1 increase; 95% CI [1.23–1.78]; $p < 0.001$) remained independently associated with poor outcomes at 90 days (Table 3). During TCCS

**Table 2** Comparison of demographic and clinical characteristics between patients with poor (mRS 3–6) and good (mRS 1–2) prognosis at 90 days after EVT.

| Variables | mRS 0–2 (n = 48) | mRS 3–6 (n = 60) | p-value[1] | p-value[2] | aOR (95% CI) |
|---|---|---|---|---|---|
| **Demographics** | | | | | |
| Age, years, median (IQR) | 64 (58–69.5) | 69 (58.8–73.8) | 0.035 | 0.388 | 1.02(0.98–1.06) |
| Male, n (%) | 36 (75.0) | 39 (65.0) | 0.262 | 0.313 | 1.61 (0.64–4.07) |
| **Medical history, n (%)** | | | | | |
| Hypertension | 30 (62.5) | 42 (70.0) | 0.411 | 0.429 | 1.43 (0.59–3.47) |
| Hyperlipidemia | 18 (37.5) | 16 (26.7) | 0.228 | 0.283 | 0.61 (0.25–1.50) |
| Diabetes mellitus | 9 (18.8) | 14 (23.3) | 0.563 | 0.721 | 0.82 (0.27–2.45) |
| Coronary artery disease | 17 (35.4) | 18 (30.0) | 0.550 | 0.942 | 1.04 (0.41–2.60) |
| Atrial fibrillation | 8 (16.7) | 19(31.7) | 0.074 | 0.180 | 2.01 (0.72–5.56) |
| Smoking | 23 (47.9) | 27 (45.0) | 0.763 | 0.865 | 1.07 (0.46–2.52) |
| Alcohol | 23 (47.9) | 26 (43.3) | 0.635 | 0.705 | 0.85 (0.37–1.97) |
| Previous stroke | 6 (12.5) | 14 (23.3) | 0.150 | 0.455 | 1.57 (0.48–5.15 ) |
| Antiplatelet therapy/oral anticoagulation | 6 (12.5) | 9 (15.0) | 0.709 | 0.415 | 1.71 (0.47–6.25) |
| **Stroke mechanism, n (%)** | | | | | |
| LAA | 28 (58.3) | 33 (55.0) | 0.728 | 0.581 | 0.79 (0.34–1.85) |
| Cardioembolism | 10 (20.8) | 20 (33.3) | 0.150 | 0.178 | 1.94 (0.74–5.12) |
| Other etiologies | 10 (20.8) | 7 (11.7) | 0.194 | 0.352 | 0.57 (0.17–1.87) |
| **Admission NIHSS, median (IQR)** | 17 (9.25–19.75) | 20 (16–23) | 0.001 | 0.023 | 1.07 (1.01–1.13) |
| **ASPECTS, median (IQR)** | 9 (8–9.75) | 8 (8–9) | 0.001 | 0.010 | 0.53 (0.32–0.86) |
| **Procedure timings** | | | | | |
| Onset-to-puncture time, min, median (IQR) | 253.5 (173–319.3) | 258 (215–304.8) | 0.383 | 0.189 | 1.00 (1.00–1.01) |
| Onset-to-reperfusion time, min, median (IQR) | 332.5 (277–421.5) | 366.5 (315–430.3) | 0.092 | 0.067 | 1.00 (1.00–1.01) |
| Reperfusion-to-TCCS time, min, median (IQR) | 255.5 (197.3–331.3) | 220.5 (168.0–291.8) | 0.058 | 0.149 | 1.00 (0.99–1.00) |
| **MBP during TCCS monitoring, mean ± SD, mmHg** | 102.2 ± 14.28 | 97.1 ± 13.28 | 0.057 | 0.066 | 0.97(0.94–1.00) |
| **Occlusion artery, n (%)** | | | | | |
| ICA+MCA | 5 (10.4) | 9 (15.0) | 0.481 | 0.775 | 1.21 (0.34–4.33) |
| ICA | 11 (22.9) | 20 (33.3) | 0.234 | 0.626 | 0.81 (0.34–1.93) |
| MCA | 32 (66.7) | 31 (51.7) | 0.116 | 0.750 | 1.17 (0.45–3.08) |
| **Acute stroke therapy, n (%)** | | | | | |
| Bridging thrombolysis | 20 (41.7) | 21 (35) | 0.478 | 0.692 | 0.84 (0.36–1.97) |
| Intra-arterial thrombolysis | 9(18.8) | 10(16.7) | 0.778 | 0.736 | 0.83 (0.27–2.49) |
| Stent retriever | 40 (83.3) | 47 (78.3) | 0.514 | 0.516 | 0.70 (0.24–2.06) |
| Aspiration | 3 (6.3) | 8 (13.3) | 0.339 | 0.568 | 1.55 (0.34–.01) |
| Angioplasty or stent | 13(27.1) | 22 (36.7) | 0.290 | 0.194 | 1.82 (0.74–4.48) |
| mTICI, n (%) | | | | | |
| 2a | 3 (6.3) | 5 (8.3) | 0.730 | 0.960 | 0.96 (0.18–5.20) |
| 2b | 3 (6.3) | 4 (6.7) | 1.000 | 0.735 | 1.34 (0.25–7.17) |
| 3 | 42 (87.5) | 51 (85.0) | 0.785 | 0.835 | 0.88 (0.26–3.01) |

**Notes.**

p- value[1] was for comparison of two groups; p- value[2] was for regression model. Variables with a p-value[1] <0.05 were selected for the multivariate analysis.

aOR, adjusted odds ratio; ASPECTS, Alberta Stroke Program Early Computed Tomography Score; CI, confidence interval; EVT, endovascular therapy; ICA, internal carotid artery; IQR, interquartile range; LAA, large artery atherosclerosis; MBP, mean blood pressure; MCA, middle cerebral artery; mRS, modified Rankin Scale; mTICI, the modified treatment in cerebral infarction; NIHSS, National Institute of Health Stroke Scale; SD, standard deviation; TCCS, transcranial color-coded sonography.

**Table 3** Comparison of post-EVT ultrasonographic findings between AIS patients with poor (mRS 3–6) and good (mRS 1–2) prognosis at 90 days.

| | mRS score 0–2 (n = 48) | mRS score 3–6 (n = 60) | p- value[1] | p-value[2] | aOR (95% CI) |
|---|---|---|---|---|---|
| PSV, mean ± SD, cm/s | 175.9 ± 74.1 | 154.1 ± 49.4 | 0.082 | 0.148 | 1.01 (1.00–1.01) |
| EDV, mean ± SD, cm/s | 60.1 ± 21.0 | 71.3 ± 33.0 | 0.083 | 0.115 | 1.01 (1.00–1.03) |
| MFV, mean ± SD, cm/s | 93.3 ± 27.5 | 98.5 ± 36.9 | 0.414 | 0.397 | 1.01 (0.99–1.02) |
| PI, mean ± SD | 1.00 ± 0.21 | 1.05 ± 0.24 | 0.356 | 0.561 | 1.73 (0.27–10.87) |
| PSV ratio, mean ± SD | 1.35 ± 0.31 | 1.60 ± 0.64 | 0.016 | 0.022* | 1.11 (1.02–1.22) |
| MFV ratio, mean ± SD | 1.22 ± 0.25 | 1.48 ± 0.64 | 0.010 | 0.007* | 1.15 (1.04–1.28) |
| Adjusted PSV ratio, median (IQR) | 1.37 (1.09–1.61) | 1.75 (1.38–1.99) | <0.001 | <0.001* | 1.31 (1.13–1.50) |
| Adjusted MFV ratio, median (IQR) | 1.24 (1.11–1.41) | 1.60 (1.27–1.86) | <0.001 | <0.001* | 1.48 (1.23–1.78) |

Notes.

p-value[1] was for comparison of two groups; p-value[2] was for regression model.

Multivariate logistic regression models for poor functional outcome were adjusted for admission NIHSS and ASPECTS. PSV ratio, adjusted PSV ratio, MFV ratio, and adjusted MFV ratio in these models are shown as every 0.1 increase, respectively.

*Significant p- values after correction of multiple comparisons.

AIS, acute ischemic stroke; aOR, adjusted odd ratio; ASPECTS, Alberta Stroke Program Early Computed Tomography Score; CI, confidence interval; EDV, end diastolic velocity; EVT, endovascular therapy; IQR, interquartile range; MFV, mean flow velocity; mRS, modified Rankin Scale; NIHSS, National Institute of Health Stroke Scale; PI, pulsatility index; PSV, peak systolic velocity; SD, standard deviation.

monitoring, no significant differences in the reperfusion-to-TCCS time and mean blood pressure were detected between two groups ($p = 0.058$ and $0.057$, respectively) (Table 2).

## ROC curves for prediction of poor functional outcomes

Table 4 displays the results of the univariate ROC analysis for PSV ratio, adjusted PSV ratio, MFV ratio and adjusted MFV ratio. The adjusted PSV ratio and adjusted MFV ratio achieved higher area under the curve (AUC) values than the others ($p < 0.05$). The adjusted PSV ratio had a sensitivity of 56.7%, a specificity of 81.3%, a positive predictive value of 79.1%, and a negative predictive value of 60.0%. The adjusted MFV ratio had a sensitivity of 58.3%, a specificity of 93.8%, a positive predictive value of 92.1%, and a negative predictive value of 64.3%. No statistically significant difference was observed in the AUC values between the adjusted PSV ratio and the adjusted MFV ratio ($p = 0.672$). Therefore, both ratios were further evaluated in the predictive models for their ability to predict poor functional outcomes in patients with AIS undergoing EVT. We used ROC curves to assess the effectiveness of the prediction models. Model A included age, admission NIHSS score, ASPECT score, and onset-to-reperfusion time and achieved an AUC of 0.75 (95% CI [0.66–0.84]) (Fig. 3). Adding the adjusted MFV ratio (Model C), but not the adjusted PSV ratio (Model B), to Model A significantly improved the predictability of 90-day poor functional outcomes compared with that in Model A ($p = 0.013$). The ROC of Model C had an AUC of 0.85 (95% CI [0.79–0.92]; Hosmer-Lemeshow goodness-of-fit test: $\chi^2 = 11.67$; $p = 0.167$), which indicated that the predictive model achieved good discrimination and calibration.

**Table 4  The different predictive values of TCCS indicators for a poor 90-day functional prognosis in AIS patients after EVT.**

|  | AUC | p-value | Cutoff value | Sensitivity % | Specificity % | Positive Predictive Value | Negative Predictive Value |
|---|---|---|---|---|---|---|---|
| PSV ratio | 0.63 | 0.015 | 1.66 | 48.33 | 81.25 | 76.31 | 55.71 |
| MFV ratio | 0.63 | 0.019 | 1.55 | 48.33 | 81.25 | 90.62 | 59.21 |
| Adjusted PSV ratio | 0.74 | <0.001 | 1.66 | 56.67 | 81.25 | 79.07 | 60.00 |
| Adjusted MFV ratio | 0.76 | <0.001 | 1.55 | 58.33 | 93.75 | 92.10 | 64.28 |

**Notes.**

AIS, acute ischemic stroke; AUC, area area under the curve; EVT, endovascular treatment; MFV, mean flow velocity; PSV, peak systolic velocity; TCCS, transcranial color-coded sonography.

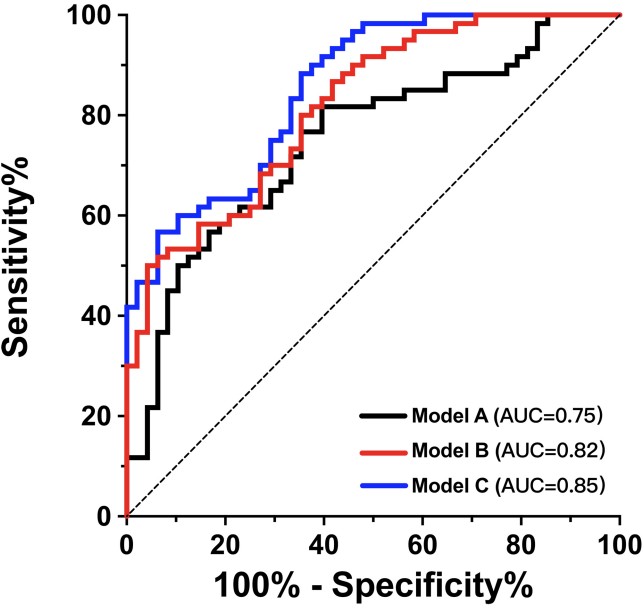

**Figure 3  Adjusted MFV ratio improved the efficacy of predicting 90-day functional outcome after EVT.** The ROC curve of model A was composed of age, admission NIHSS score, ASPECTS and onset-to-reperfusion time to predict mRS score 3–6 at 90 days. Adding adjusted MFV ratio to model A as model C significantly improved the prediction ability (AUC improved from 0.75 to 0.85, $p = 0.013$). ASPECTS, Alberta Stroke Program Early Computed Tomography Scores; AUC, area under the curve; EVT, endovascular therapy; MFV, mean flow velocity ratio; mRS, modified Rankin Scale;NIHSS, National Institutes of Health Stroke Scale; ROC, receiver operating characteristic.

## DISCUSSION

In this study, we extensively investigated the potential of TCCS indicators in predicting the prognosis of patients undergoing EVT for acute LAO in the anterior circulation. Our findings revealed that during early reperfusion, PSV, EDV, MFV, and PI were significantly elevated in the iMCA compared with in the cMCA. Furthermore, higher PSV ratio, adjusted PSV ratio, MFV ratio, and adjusted MFV ratio were found to be independent predictors of poor functional outcomes (mRS: 3–6) at 90 days, and the ROC analysis indicated that higher aPSV ratio and aMFV ratio showed better prediction ability for poor prognosis.

Finally, adding the adjusted MFV ratio significantly improved the prediction performance of the basic model for the 90-day poor functional outcomes. Although larger studies are required for further confirmation, these findings suggested that TCCS can play a larger role in the prognostic assessment of patients with ischaemic stroke.

Some randomize clinical trials have demonstrated the efficacy of EVT in patients with AIS caused by a proximal intracranial occlusion within the anterior circulation (*Kneihsl et al., 2018*). With the increasing therapeutic experience and technical improvements, successful recanalization can now be achieved in up to 90% of all thrombectomy cases (*Saver et al., 2015*); however, only 50–60% of patients show favourable outcomes (*Goyal et al., 2016*). This highlights that factors other than simple arterial recanalization may play a role in the clinical recovery of patients with AIS (*Baracchini et al., 2019a*). Cerebral autoregulation impairment, residual stenosis of the culprit arteries, and systemic hyper- or hypo-tension contribute to haemodynamic abnormalities that result in unfavourable functional outcomes (*Krishnan, Mays & Elijovich, 2021*; *Nogueira et al., 2022*; *Salinet, Panerai & Robinson, 2014*).

Previous studies have reported a significant increase in the BFV value in the treated arteries after undergoing EVT (*Baracchini et al., 2019a*; *He et al., 2020*; *Kneihsl et al., 2022*). One study evaluated the haemodynamics immediately after mechanical thrombectomy and revealed that the mean PSV reached 278.9 cm/s (*Baracchini et al., 2019a*). We obtained similar findings; however, the mean PSV in the treated MCA was only 166.2 cm/s. This discrepancy might be attributed to differences in the timing of the ultrasound monitoring. These findings indicated that hyperperfusion frequently occurs after successful recanalization.

Several studies have investigated the role of TCCS indicators in predicting clinical outcomes in patients undergoing EVT. *Kneihsl et al. (2018)* found that a high MFV ratio within 24 h of successful recanalization for AIS was predictive of ICH and poorer functional outcomes. However, other studies have reported inconsistent results. For instance, in another study, 27 patients experienced an acceleration of segmental blood flow in the iMCA (35–40% higher than that in the cMCA) within 7 days after recanalization; none showed clinical deterioration (*Perren et al., 2018*).

Consistent with previous research, our study revealed that high PSV ratio and MFV ratio were independent and comparable predictors of poor 90-day outcomes in patients with AIS undergoing EVT. Pathophysiological reasons for the high BFV value on the ipsilateral side after undergoing EVT may include hyperperfusion, residual stenosis, luminal narrowing caused by intimal hyperplasia, or transient vasospasms (*Aoki et al., 2013*; *Baracchini et al., 2019a*; *Baracchini et al., 2019b*; *Kneihsl et al., 2018*). Reactive hyperperfusion is thought to be associated with vasogenic oedema and haemorrhagic transformation (*Krishnan, Mays & Elijovich, 2021*), whereas residual stenosis and vasospasm may contribute to the enlargement of the infarct size (*Saqqur et al., 2007*). All of the above conditions lead to a poor prognosis.

Although most patients in our study had a higher BFV value in the iMCA than in the cMCA, a small proportion experienced a decrease in the BFV value. In patients with poor vascular access, the blood flow at the stenosis site usually showed turbulence and eddies;

however, blood flow deceleration at the distal end of the stenosis was attainable (*Chen et al., 2020*). Thus, residual intracranial stenosis of the ICA may decrease the BFV value in the MCA. Furthermore, even with successful recanalization, reocclusion of the target vessel occurred in 3% of patients (*Millan et al., 2017*). All these factors may lead to adverse functional outcomes and an increased risk of mortality (*Baracchini et al., 2019a*; *Chen et al., 2020*). In addition, lowering blood pressure using drugs such as antihypertensives, sedatives, and analgesics; or due to heart failure may exacerbate the decrease in the BFV value and aggravate cerebral ischaemia.

Recognising that either too high or too low a BFV value in the iMAC might cause adverse effects in patients with AIS after undergoing EVT, we introduced the adjusted BFV ratio that not only reduced individual differences but also reflected the BFV variations in the recanalised and contralateral arteries. Our findings showed that an adjusted PSV ratio of ≥ 1.66 and an adjusted MFV ratio of ≥ 1.55 were independent predictors for poor functional outcome, and their prediction ability for poor prognosis was stronger than that of PSV ratio and MFV ratio. Our results suggested that significant differences in PSV or MFV between the iMCA and cMCA might indicate poor patient prognosis. However, it is noteworthy that the BFV of the iMCA during the initial stage of recanalization might be unstable. *Baracchini et al. (2019a)* reported that the mean iPSV increased immediately after undergoing EVT; however, it gradually decreased and stabilised 1 week after the procedure, suggesting that the optimal cut-off point for the BFV value is time-limited, and differences in the timing of ultrasound monitoring might lead to discrepancies between various studies. However, we chose an earlier monitoring time because we believe that earlier identification of the pathological BFV ratio guides lifesaving therapies and improves neurological outcomes in the early stages. In our study, the median time interval between EVT and TCCS was 3.8 h.

The MFV provides a more comprehensive measurement value throughout the perfusion cycle than the PSV that only captures the maximum flow velocity during systole. Therefore, the adjusted MFV ratio was a more accurate indicator of haemodynamic disparities between the ipsilateral and contralateral MCA throughout the perfusion period than was the adjusted PSV ratio. We found that the prediction efficacy of 90-day poor functional outcomes improved after adding the adjusted MFV ratio to the basic prediction model according to the ROC analysis, suggesting that the MFV ratio is not only a valuable prognostic indicator but is also independent of traditional clinical indicators, such as admission NIHSS and ASPECT scores. Thus, this study provides a meaningful additional parameter to the growing literature, indicating the predictive ability in assessing the prognosis of AIS.

However, this study had several limitations. First, the generalisability of our results might be limited because of poor or closed temporal window penetration. Second, due to the unavailability of MRA, CTA, or DSA examination results within three months after EVT, we were unable to determine the specific pathological mechanisms that contributed to the observed hemodynamic differences during the follow-up period. Third, our study only included one ultrasound monitoring session after EVT. Monitoring at different time points would have provided more accurate information for selecting an optimal observation time. Fourth, our research findings were based on data obtained from a single centre; thus, they might not be representative of the wider population. Therefore, further large-scale

multicentre studies are required to confirm the predictive role of haemodynamic findings following anterior circulation stroke recanalization.

## CONCLUSIONS

This study demonstrated that post-EVT ultrasound monitoring in patients with stroke is a useful bedside method for identifying patients with poor outcomes, indicating that TCCS could be used as a real-time tool for better post-thrombectomy management in patients with AIS. The novel TCCS parameter, adjusted MFV ratio, can be combined with conventional parameters to provide a better and more accurate model that predicts the outcomes of patients with cerebral vessel recanalization.

### Funding

The study was supported by grants from the Science and the Technology Development Project of Tai'an (No. 2021NS203). The funders had no role in study design, data collection and analysis, decision to publish, or preparation of the manuscript.

### Grant Disclosures

The following grant information was disclosed by the authors:
Science and the Technology Development Project of Tai'an:  2021NS203.

### Competing Interests

The authors declare there are no competing interests.

### Author Contributions

- Yanyan Hu analyzed the data, prepared figures and/or tables, authored or reviewed drafts of the article, and approved the final draft.
- Shizhong Zhang conceived and designed the experiments, authored or reviewed drafts of the article, and approved the final draft.
- Jiajun Zhang conceived and designed the experiments, prepared figures and/or tables, authored or reviewed drafts of the article, and approved the final draft.
- Xin Wang performed the experiments, authored or reviewed drafts of the article, and approved the final draft.
- Feng Zhang performed the experiments, authored or reviewed drafts of the article, and approved the final draft.
- Hong Cui performed the experiments, authored or reviewed drafts of the article, and approved the final draft.
- Hui Yuan conceived and designed the experiments, analyzed the data, authored or reviewed drafts of the article, and approved the final draft.
- Wei Zheng conceived and designed the experiments, performed the experiments, analyzed the data, prepared figures and/or tables, authored or reviewed drafts of the article, and approved the final draft.

## Human Ethics

The following information was supplied relating to ethical approvals (i.e., approving body and any reference numbers):

The Second Affiliation Hospital of Shandong First Medical University granted Ethical approval to carry out the study within its facilities (2021-098).

## Ethics

The following information was supplied relating to ethical approvals (i.e., approving body and any reference numbers):

The Second Affiliation Hospital of Shandong First Medical University approved the study (2021-098).

## Data Availability

The raw data is available as a Supplementary File.

## Supplemental Information

Supplemental information for this article can be found online at http://dx.doi.org/10.7717/peerj.15872#supplemental-information.

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
