# Peer review of "Early haemodynamic predictors of poor functional outcomes in patients with acute ischaemic stroke receiving endovascular therapy: a single-centre retrospective study in China"

_PeerJ, doi:10.7717/peerj.15872_

## Round 0.1 · original submission · Minor Revisions

Please address the reviewers' comments.

Reviewer 1 ·

Basic reporting

1.The quality of the English language used in the text is insufficient and requires correction of grammar errors, such as in line 45 where 'adult patients' is repeated. 
2.Did the authors obtain consent from the deceased patient's family members or the need for informed consent was waived?

Experimental design

1. Were postoperative complications, such as enlarged infarct size, intracranial hemorrhage, and cerebral edema, monitored and recorded by the authors? If so, please provide the relevant data. If not, this point should be added as a limit.
2. The method should be adequately described with enough information, such as the gain and depth setting of transcranial color-coded sonography.

Validity of the findings

1. The study conducted by the author suggests that the adjusted mean flow velocity ratio is a better predictor of poor prognosis than the adjusted the peak systolic velocity ratio. It would be helpful if the author could provide an explanation for the observed results from the perspective of hemodynamics.
2. Performing ultrasound monitoring only once and selecting the monitoring time point at random may lead to concerns regarding the reliability of the obtained results.

Additional comments

None.

Reviewer 2 ·

Basic reporting

Language and expression should be improved.
1. In Methods (Line 45), the authors should describe the method as the name indicates. For instance: Retrospective study in a single academic hospital.
2. In Results the authors should use univariable logistic regression analysis instead of “univariate analyses” (Line 181) and they also should use multivariable logistic regression analysis instead of “logistic regression analysis” (Line 55, 184).
3. “MATERIALS AND METHODS” in Line 89 should be changed to "Survey methodology".
4. “mTICI grades 3” in Line 168 should be changed to “mTICI grade 3”.
Furthermore, there are many punctuation, spacing, and format errors. It is recommended that the authors seek the assistance of an expert in English editing and academic writing to improve the clarity, flow, and format of their manuscript.

Experimental design

1. Did the authors used ultrasound contrast agents in their study? If so, disclose the name of the contrast medium, dosage, or manufacturer. If not, explicitly state in the article.
2. The inclusion of patients with partial recanalization in the literature is often questioned. Please explain why select patients with partial recanalization, and provide a rationale.
3. The study did not provide clear pathological reasons for the observed changes in blood flow velocity after EVT surgery. Did the patients undergo DSA, MRA, or CTA imaging after the surgery to identify the underlying cause of these changes?

Validity of the findings

The Discussion section of the manuscript needs to be revised. It should start with a clear statement of the main finding(s) of the study. Following this, comments related to the present findings should be discussed in relation to prior work in the literature. It is important to support each sentence related to comments on prior work with no more than one reference. Additionally, the strengths and limitations of the study should be addressed, followed by a clear statement of the main conclusion(s) of the study.

Additional comments

The reference list requires formatting to adhere to the journal style.

Reviewer 3 ·

Basic reporting

The authors of the present paper aimed to clarify the association between hemodynamic indicators and functional outcome at 90 days after EVT, and evaluated their predictive usefulness. While the content is adequate and interesting, the wording could be improved. It is recommended that
1.English editing throughout should be considered.
2.The article should include sufficient introduction and background to demonstrate how the work fits into the broader field of knowledge.
3.To ensure consistency and clarity, it is recommended to use a consistent expression for follow-up time throughout the manuscript. In some places, ‘90 days’ is used while in others ‘3 months’ is used. It is suggested to choose one expression and use it consistently throughout the manuscript.

Experimental design

1.The study did not provide information on deaths from non-CNS causes among patients with poor prognosis. It is unclear whether the authors excluded such patients or not. Please provide the above information during revision of your manuscript.
2.The duration of this study is only one year, why not conduct a longer study to collect more case data.
3.Please describe the inclusion criteria of research subject clearly, as only the exclusion criteria have been explained so far.

Validity of the findings

The authors' intentions are commendable and their work is focused on providing a comprehensive understanding of the hemodynamic changes that occur after EVT. This knowledge can potentially aid physicians in making informed clinical decisions. However, the study did not identify the exact pathological mechanism such as hyperperfusion, local vasospasm, or vascular stenosis, that led to the patient's hemodynamic changes.

---

## Round 0.2 · accepted · Accept

Thanks to the authors, reviewers, and editors. This is a good revised manuscript, that can be accepted.

Reviewer 2 ·

Basic reporting

The author's article has been greatly improved.

Experimental design

The design of the article is reasonable.

Validity of the findings

The conclusion of the article is reliable.

Additional comments

I think the article should be accepted.

Reviewer 3 ·

Basic reporting

no comment

Experimental design

no comment

Validity of the findings

no comment

Additional comments

no comment